# Predictive Sequential Research Design to Study Complex Social Phenomena

**DOI:** 10.3390/e23050627

**Published:** 2021-05-18

**Authors:** Romel Ramón González-Díaz, Gladys Inés Bustamante-Cabrera

**Affiliations:** 1Centro Internacional de Investigación y Desarrollo—CIID, Monteria 230001, Colombia; 2Latin American Bioethics Committee of the Centro Internacional de Investigación y Desarrollo—CIID, Monteria 230001, Colombia; dra.gbustamante@gmail.com

**Keywords:** mixed methodology, research methods, social sciences, complex thinking, design, artificial intelligence, diffuse logic, pluralism, qualitative research

## Abstract

Social phenomena in their simplest form share infinite complexities and relationships, and by interacting with other entities, their levels of complexity become exponentially inexplicable and incomprehensible. Using a single form of study in complex phenomena could be insufficient, and new forms of analysis should be opened that allow for observing the multidimensionality of study problems from integrative perspectives. The emergence of research using mixed methods attempts to reconcile these methodologies through integration, configuring a stage of interconnection between research paradigms that cause cuts and leaks that may or may not be consistent with the study’s object. At the time of integration, vices can be created by specific value and subjectivity judgments, with investigative diffraction being an alternative to extend integration through data fracture and redirecting the object of study. This work proposes a Predictive Sequential Research Design (DISPRE) for complex social phenomena, which uses fuzzy logic as a tool to solve the information biases caused by the investigative diffraction of each methodological approach as a strategy to capture, explain, understand and predict the intrinsic complexity of the social entity under study.

## 1. Introduction

Theoretical, epistemic and ontological debates are at the centre of critical debates when taking a methodological stance (quantitative or qualitative) in the social sciences. However, these gaps have now been overcome through paradigmatic complementarity and integration for knowledge generation [1].

New methodological approaches aim to account for science holistically for those study phenomena that involve higher levels of complexity through epistemic pluralism [2]. Pragmatism is the philosophical connection of mixed methods, based on the worldview of actions, situations, and consequences, rather than antecedent conditions (as in post-positivism), which is concerned with applications (what works) and solutions to problems. 

Traditional methods, considered as “normal” and “coherent”, ignore this qualitative and quantitative connection, as well as the methodological bridges, which are appropriate for these novel and integral methods, establishing instead bridges of a social nature that seek to provide an integral reason for the events. A mixed methodological multilevel [3] approach could explain or understand this object of study through complementarity [4] as a principle of complex thinking [5], manifested in the emerging social phenomena, in such a way that the absence of replicability and extensibility raised in the logic of the formal process is explained in Morin’s hologrammatism, as a representable generalisation which goes beyond the holistic, establishing dynamic, flexible research which overlaps with the rigid knowledge of classical methodologies [6].

These new research perspectives cover the social sciences, characterised by the inexplicable and incomprehensible conception of the study of reality under a single research perspective, revealing the existence of complex phenomena that require systematic processes that combine inductive and deductive logic at any stage of the research [7], which allows the study of the social phenomenon with greater breadth, to cover all its complexity and give support to multi-method studies, also known by their other names: multiple, mixed, integrated, multimodal or multi-strategy [4,8]. However, this method’s selection must be argued in its procedure and techniques of data collection, processing, and analysis to give it legitimacy [4] and generate sufficient data to interpret the dimensions and meanings from different aspects globally.

Thus, mixed methods can be applied in most branches of knowledge, meaning that their application in health sciences and education is not only possible but also valid, because this type of study allows us to understand the problem of the representation of specific clinical or philosophical-epistemological entities from the point of view of social phenomena and vice versa, proposing a more comprehensive analysis of a particular subject, whose background lies in the pragmatism of the researcher and the judicious use of methodological triangulation of different types of information for the same question, in such a way that their instrumentalisation described in the contributions of Denzin and Lincoln [9] and Morse and Chung [10] has been considered as diachronic or synchronic strategies in time [11,12], since they reduce biases in the use of compensatory analytics. It is therefore rational to understand that qualitative methods, which also give validity to mixed studies, are integrated with quantitative ones, when the procedure of analysis of complex social phenomena deriving in disease processes or vice versa is intuited, or when similar educational conditions exist in disparate societies, within the same geographical context.

In this way, mixed studies use different sources of information, which are combined according to the researcher’s objectives to achieve a more comprehensive basis for analysis, using analytical logic as a substantial basis for the procedure.

In this sense, mixed methods research generates nuanced and comprehensive understandings through the use of different methodological tools to answer complex research questions [13], which is a challenge for researchers, who must creatively and logically design and implement the combination of methods to generate meaningful and defensible conclusions [14]. It is understandable, then, that two or more methods may intentionally be used to evaluate the same conceptual phenomenon to preserve countervailing biases. It is indirectly emphasised that Cook’s multiplism that emphasises validity enhanced by the convergence of results from multiple methods is more complex and approaches the triangulation of mixed methods, empowering this method’s validity as a complex alternative in research processes [15]. This perception forces a reconceptualisation of knowledge, personhood and human society, underpinned by the pragmatic principle that the only truth lies in utility: usefulness supersedes correctness [16].

The scientific literature shows the scarce use of mixed methods for solving problems of social phenomena. The existing publications denote a confusing research design and a low level of rigour in the application of mixed methods (Harrison et al., 2020), which raises doubts regarding the methodological quality from the perspective of Hong and Pluye (2019), who relates it to the construction of reliability, evaluation of procedural aspects and the accuracy in which the results account for the research question.

It is mentioned that mixed methods have five purposes to be fulfilled: (a) triangulation; (b) complementarity; (c) development; (d) initiation; and (e) expansion, whether of methods, phenomena, paradigms, status or procedural implementation in a similar, different, equal, interactive, independent, sequential or concurrent manner [17]. Methodological debates on mixed methods research focus on their typology and application, making essential contributions to exploring and explaining new social sciences findings. However, they are still insufficient if they do not generate the necessary elements for decision making; despite this, Gallaudet University in the USA has shown substantial growth in this type of study, offering a theoretical efficiency “that allows robust, multidimensional analysis of social phenomena, much more effective than when using a single method [18].

Motivated by this situation, the researchers propose a mixed methodology for the study of complex phenomena in the social and humanistic sciences, as well as the generation of novel creations that promote more competitive and innovative funding: the Sequential Predictive Research Design (DISPRE) is a novel reference that aims to minimise biases in the research process and confers, through fuzzy logic, a predictive model based on the dynamics of the input variables to the inference engine (knowledge base and rules) and the retro-prospective study of the social entities under study. Therefore, this study is presented in three phases: conceptualisation and mixed methods, description of DISPRE, application proposals and future developments or limitations.

## 2. Mixed Methods in the Social Sciences

The definitions of mixed methods are diverse and have evolved, as evidenced by the theoretical discussions of Johnson et al. (2007). After reviewing the scientific publications of Patton (2014) and Hunter and Brewer (2015), they concluded that mixed methods studies represent a theoretical and practical synthesis based on quantitative and qualitative research, constituting the third research paradigm that provides broader and more valuable results. This method, considered daring by some researchers because of the eclecticism of the methodology, the relevance of the research questions and the use of plural paradigms, is neither a limitation nor a threat to classical research methods, as mixed methods are associated with pragmatism and any logic that can contribute to the construction of defensible research results.

This innovative mode of study is only helpful in situations where the research questions are complex and intended to address phenomena of the same range, with interdisciplinary participation [19], with different theoretical and practical perspectives, encouraging fuzzy thinking and logic. This complementarity, not seen before in other types of studies, establishes particular correlations and critical positions and formalisations in the methodological combination that gives it an innovative and futuristic character.

This methodology, which has been documented since 1920 [20] in anthropology and psychology, or in the use of triangulation and convergent validation and multioperationalism [21] or in the production of the works of Denzin and Lincoln [9], led us to the remembrance that these types of methodologies were already applied empirically, but futuristically, avoiding unnecessary disqualifications and exclusions between methods. Similarly, the relationship between work, health or educational and occupational strategies has already been studied in Argentina by Gallart and in Spain by Valisachis de Gialdino [22,23].

Consequently, the social sciences dedicated to studying the complexity of organisations have been undergoing substantial changes in the use of methodologies for the generation of scientific knowledge and support for decision-making. In this sense, McKim [24] described a series of studies conducted in the organisational field, with the result being that in recent years, the value in the use of mixed methods has been increasing the validity of the findings, informing the collection of the second source of data, and helping with the creation of knowledge. Other studies by Fetters and Molina-Azorin [25] found that articles on mixed methods applied to the social sciences, especially in education, were more cited than those that did not use them.

In this sense, Morse and Chung [10] and Tashakkori and Creswell [26] agreed that social science studies conducted using mixed methods provide greater certainty in research results. According to O’Cathain [27], this robustness is derived from the integration as the core of mixed-methods exercise due to the cross-referencing of multiple sources of information. On the other hand, following Uprichard and Dawney [6], who outlined critical debates about integrative positions, we argue that all mixed methods research generates “cuts” that may or may not be coherent. That diffraction can assume a broadening of integration due to the fracturing of data that could occur in the research process and distort the object of study [28], presenting in the social sciences a way of capturing the complexity intrinsic to the ontology of the social entity being studied.

The human being appropriates information with which they establish their ideas in a particular way, applying logic in a differentiated way depending on the events analysed, in such a way that classical logic may not be enough to explain some critical reasoning, as happens in mixed research, which involves unconventional categories at the time of their interpretation. In turn, fuzzy logic through data analysis helps us to understand some of this complexity due to fuzzy set theory’s benefits that offer practical tools to characterise human expert knowledge [29]. Within fuzzy logic, a theory is Fuzzy Rule Base Systems, which can be explained as a system of rules in which fuzzy sets in combination with fuzzy logic are used to describe expert knowledge based on the objective of the study and to model the relationships between input and output variables to overcome knowledge uncertainty [30].

Algorithms for analysing multi-criteria decision making and social choice in situations plagued by inaccuracies are evidenced in the scientific literature, paying close conceptual attention to fuzzy preference relationships and formulating data aggregation tools based on fuzzy social choices. For this, we used the framework proposed by Arfi [31] that works exclusively with linguistic fuzzy-logical methods in the social sciences and the framework proposed by Hernández-Julio et al. [32].

### 2.1. Mixed Methods Research Designs

In mixed methods, the following three types of research design can be grouped as techno-methodological strategies [3], as interpreted by [4].

#### 2.1.1. Convergent Designs

Alternatively called “triangulation” or “parallel” designs, convergent designs involve a confrontation of data from different (quantitative and qualitative) perspectives in order to merge them, confront them and generate a comprehensive interpretation with a holistic view of the object of study [8] (See Figure 1). Triangulation increases confidence, overcomes the reductionism of approaches, and, if there are congruencies in the conclusions, confers the results’ reliability and validity [4]. A convergent model can be considered when researchers need to use qualitative and quantitative data concurrently to compare results. Thus, in an experimental-type study, a qualitative element is added to understand the intervention procedure [33].

These designs can lead to frequency, factor or outcome comparisons and their analysis is consolidated when both sets of data are collected and analysed separately, leading to meta-inferences for the integration of information [34]. Thus, in this method, four forms of triangulation can be stated:-Triangulation of data considers temporal and spatial dimensions and subjects for comparisons. For this, the methods for the interpretation of the phenomenon must be qualitative to be comparable. In this case, information is verified at different times and by different methods. It is possible to find differences in information that do not diminish the credibility of the information but of the source of information when the phenomenon occurred.-Researcher triangulation refers to the different perspectives of the research subjects on the same study phenomenon to minimise the interpretative error of the researcher’s value judgments. This is usually done by taking into account people from different disciplines, reducing the bias of using one researcher exclusively. Data analysis will also be done individually by each researcher and finally subjected to comparative analysis of results.-Theoretical triangulation refers to the study of the phenomenon from different theoretical positions; the conception of the theory to be used during the analysis of the research must be chosen beforehand so that when the comparison is made, the difference of premises with the same group of data or information becomes evident.-Methodological triangulation attempts to generate convergence through different methodological perspectives. The importance of this type of procedure focuses on the ability to elucidate the complementary parts of a whole phenomenon and identify the reasons why different methods offer different results [35,36].

The process of methodological triangulation in mixed methods research can be generated in two ways: (1) intra-method triangulation, wherefrom a methodological component, the reliability and validity of the information is reinforced with the application of different techniques and strategies, which is in line with the intra-paradigmatic multi-method approach; and (2) inter-method triangulation, which refers to the convergence of methodologies, each with its dominant method to triangulate the results as complementation, also known as an inter-paradigmatic multi-method approach [34]. Both processes of analysis relate to criterion validity, where the results of each collection technique and instrument are contrasted with each other and related, conferring the desired robustness.

#### 2.1.2. Explanatory Sequential Designs (DEXPLIS)

This design is also called “concurrent and embedded of dominance (dominant: Quantitative)”, with technical-operational strategies of combination. It is described as a procedure of the quantitative (dominant) component, which generates some results, and these are deepened with the qualitative (complementary) component [37] (see Figure 2).

Explanatory sequential designs use the quantitative (dominant) component’s strengths to generate statistical results, then proceed to determine the critical points of the neuralgic findings to be strengthened from a qualitative perspective [6] in such a way that the quantitative component establishes the critical links with the qualitative component for a comprehensive interpretation, giving rise to explanatory results of the phenomenon under study.

#### 2.1.3. Exploratory Sequential Designs (DEXPLOS)

This design considers the qualitative (dominant) and quantitative (complementary) components. This design’s nature lies in a prior exploration of the social phenomenon through the techniques and strategies of the dominant method, thereby determining the critical categories to delimit the object of study and proceed with a quantitative deployment. This type of design is often used for studies of unspecific problems with unknown variables [3] (see Figure 3). An advantage of this type of design is the definition of the stages, which are clear and differentiated. However, an obvious disadvantage is that the qualitative stage must be carried out by a researcher with extensive knowledge of the phenomenon, identifying the emerging categories for an adequate conceptualisation [38].

Currently, the social sciences are in a state of revaluation of social reality as a complex association that can be systematized through quantifiable and non-quantifiable dimensions in specific periods, as defined by Hashimoto and Saavedra [40], who, using different methods of techno-operational data analysis, achieve consistent results. In this sense, Blanco and Pirela [4] agreed that epistemological and methodological pluralism leads to more effective research and makes it increasingly interdisciplinary, complex and dynamic. The study of complex problems in the social sciences requires the complementarity of research approaches to provide a solid understanding of the phenomenon. However, explaining/understanding the object of study is insufficient to strategically anticipate the social entity, defined by González-Díaz and Becerra-Perez [41] as a war where anticipation is the defining strategy for organisational success. The mixed-methods described in the scientific literature implicitly describe research levels (sequential exploratory and explanatory), alluding to a combination of techno-methodological strategies detailed in the previous sections, leaving a conceptual and methodological vacuum of sequential designs that aim to predict complex phenomena.

## 3. Proposal on Sequential Predictive Research Design—DISPRE

The sequential exploratory approach has been the dominant type of design employed in the social sciences [42]. This type of study requires a research question underpinned by a firm theoretical basis, with a consensual and rational basis. The process to achieve this is the conception of prospective thinking, strategic action, and ownership to model pessimistic and optimistic scenarios to facilitate decision-making, with the generation of valuable frameworks and alternative actions chosen according to the researcher’s expertise [43]. Thus, the present study aims to design a predictive model about a complex social phenomenon’s different behaviours through fuzzy inference modules. The main idea is to address the different possible organisational scenarios. To this end, we start with an exploratory analysis of trends and, additionally, we resort to the generation of synthetic data (Monte Carlo simulation) to determine possible futures (futuribles) based on the analysis of actors and scenarios (see Figure 4).

For a better understanding of this research design, the following four phases are described below:

### 3.1. PHASE 1: Qualitative Analysis (Diagnosis)

From a qualitative perspective, the aim is to explore the study phenomenon through the inductive method, determining through hermeneutic analysis the emerging categories of the complex reality studied. To achieve this, the application of semi-structured interviews is recommended through a script of questions to key informants selected according to pre-established criteria and coherent with the object of study [46].

On the other hand, aspects of credibility must be considered, related to internal validity, which is related to how the key informants understand the object of study and the extent to which the researchers acquire this information, especially their experiences and perception of what has been studied, which is strengthened to the extent that it is subjected to discussion with peers or colleagues, taking into account the relationships established by the researchers and the interpretation, generating a process of comparison from multiple sources. Of course, the researcher’s ethical component will determine the quality of the study’s content.

In order to avoid ambiguities or biases in data collection, the instruments must be reviewed and endorsed by an accredited research ethics committee, whose requirement will be, in addition to the sustained validity of the study, the reliability of the instrument drawn up by the researcher, through the methodological approach that certifies its usefulness. The ethical criteria of qualitative research are based on the explanation of the researcher’s interpretative nature and the need to give meaning to the expressions of the subjects based on the quality of the expressions of the events [47].

In this way, the analysis of the findings can be supported by the approaches of specific processes that can reinforce the validity and reliability of qualitative studies [48], such as the following:-Categorisation: In this process, significant classes are constructed, drawing and blurring the whole and its particulars, while the information is deepened, using underlying meanings to make sense of what is being examined. Its organisation can be deductive, using theoretical references with which it defines the categories and subcategories to be studied, or inductive, in which the researcher gives birth to the information to be extracted according to the diagnosis.-Structuring: In this process, various shadows emerge, which then determine their integrating structure.-Contrasting: This stage contrasts the informants’ perceptions regarding the object of study and the theoretical structures that expand the categories and specify the study’s object.-Theorisation: This process consolidates the results with the different categories that emerge from the interview about the object of study so that the various theoretical approaches logically complement it.

To generate a greater understanding of the explored problem, DISPRE requires multiple convergences of perspectives; in other words, a triangulation of data between informants (1 and 2) (see Figure 4). Subsequently, a convergence of perspectives will allow for the generation of emerging categories.

### 3.2. PHASE 2: Integral Analysis of Inter-Paradigmatic Connection

Once the hermeneutic analysis described in the previous phase has been elaborated, categories emerge that represent the study’s problem from the key informants’ perceptions. In this phase, the discussion of three experts selected according to the following criteria: the time of experience and professional training in the area of study is used to determine the critical dimensions and configure the systematisation of the quantitative research, i.e., a critical analysis is carried out through a focus group to generate the inter-paradigmatic connection from the qualitative to the quantitative vision as a compliment.

This analysis aims to reconcile the divergences between the research approaches, using the experts’ expertise in determining the critical dimensions (quantitative), supported by the qualitative findings. For this procedure, the experts will use structural analysis as a tool for collective reflection. A weighting system is applied considering the degree of dependence and influence between categories.

Dhir and Dhir [49] agreed that structural analysis is a tool for understanding and describing different situations from a systemic and complex perspective in order to illustrate through the structuring of ideas the system in which it is immersed, based on a matrix that relates all its elements, opting for pluralism and complementarity of approaches, to reduce complexity and be able to explore multiple and uncertain futures through collective and collaborative reflection.

The tool used for the structural analysis is the Matrix of Cross Impacts-Multiplication Applied to a Classification (MICMAC) method, defined according to Leśniak and Górka [50], as a method that “consists of raising the structural analysis matrix to a power of successive values. In this way, thousands and millions of lines are analysed in most concrete systems” (p. 13). As a summary, the stages for the analysis are described.

-Stage 1—List of dimensions: the dimensions studied are listed.-Stage 2—Description of the dimensions’ relationships: the structural matrix is elaborated where each dimension must be found at a crossroads with each other dimension. The experts are then asked: is there a relationship of influence between the dimensions, according to the following weighting: no relationship = zero (0), direct influence is weak = one (1), direct influence is medium = two (2), direct influence is strong (3) or potential (P)?

Identification of the critical dimensions: this consists of analysing the direct influences and the intensity by means of qualitative assessments to determine the typology of the dimensions and their interpretation.

### 3.3. PHASE 3: Quantitative Analysis (Completion)

Once the critical dimensions have been determined through the inter-paradigmatic connection analysis, they are operationalised, using the conceptual, theoretical structures specific to each dimension and configuring the indicators required to respond to them. Likewise, the aspects of validation and reliability must be considered to generalise the results through descriptive statistics.

From a quantitative point of view, the following data collection technique was selected: the survey and the questionnaire as an instrument, preferably with a Likert-type scale (ordinal-scale), considering Tutz [51]. The questionnaire should be adjusted for each sample group (described and calculated according to the respective techniques with a confidence level of 95% and a margin of error of 5%) to collect data from reality within a natural context where the research is carried out. The instrument used must be subjected to criteria of validity and reliability following the instrument used. In the case of a questionnaire, the process of content validation, construct or discriminant validity and reliability through the Cronbach’s Alpha coefficient (when the instrument has several response alternatives and when it is dichotomous, Kuder–Richardson is used) between 0.80 and 0.95. Once the questionnaires have been applied, the data are processed to obtain information through individualised descriptive analysis of each indicator, determining the values (minimum, arithmetic mean, standard deviation and maximum).

### 3.4. PHASE 4: Predictive Modeling

For the development of the predictive model through the sequential research design, it is necessary to understand that social reality is highly complex and presents numerous edges that give it an amorphous configuration, resulting in an actual psycho-biosocial framework in its nature. Each entity’s internal and external factors intervene in a multiplicity of factors that go beyond a technical, scientific or hermeneutic proposition [52].

The configuration of the organisational predictive model considers two parallel phases before arriving at its construction. On the one hand, the so-called input variable (phase A) consists of data processing through clustering or grouping where those indicators with the most significant impact on the phenomenon under study are defined. Subsequently, they are subjected to 100,000 iterations using the Monte Carlo method to determine their impact on the study object.

On the other hand, the output variables (phase B) begin with a structural analysis (retro-prospective), allowing for the organisation’s environment’s main trends under study. Then, the key actors are analysed about the organisational objectives using the Alliances and Conflicts Matrix: Tactics, Objectives and Recommendations (MACTOR), building the respective information for the analysis of strategies with Systems and Cross-Impact Matrices (SMIC), determining the possible strategies (for more information on both analysis techniques, refer to Arango Mo-rales and Cuevas Pérez [43] (see Figure 5)).

As shown in Figure 5, the input variables (phase A) and output variables (phase B) are constructed. With these inputs, the fuzzy inference module is configured to generate the prediction algorithm.

### 3.5. Fuzzy Logic

To describe the fuzzy model, it is necessary to take into account that empirical descriptions are made “crisp” or “fuzzy” through methods that make them visible and allow for exposing or hiding the fractured, fragmented and disordered research object; evidence of these social behaviours is mentioned in the systemic instabilities described by Helbing et al. [53] and M. Perc [54]. With this in mind, the diffractive perspective through the methodology proposed by Hernández-Julio et al. [55] and applied in Hernández-Julio et al. [56] will allow us to discover the emerging realities of the methodological cut and to grant the dispersions on the research object. Hernández-Julio, Prieto-Guevara et al. [57], presented a framework for the development of decision support systems based on fuzzy sets using clusters and dynamic tables. This framework consists of the application of 11 steps (Figure 6).

The first three steps (1–3) help us to better understand the specific domain and possible gaps in existing models for evaluation. The following steps belong to the levels before the iterative design and development of data-driven decision support systems (steps 4–5). The following steps contain the iterative design and construction of data-driven decision support systems (steps 6–9). Steps 7 and 8 consist of the creation of the knowledge database. In the next step (step 9), pivot tables explain the use of this resource as a feature selector and the creation of the knowledge rule base, and the following steps (steps 10–11) concern the implementation and evaluation of Mamdani-type fuzzy data-driven decision support systems. The last step is the communication process. Through the mentioned framework of the study and that proposed by Arfi [31], Abadeh et al. [56], working with linguistic fuzzy-logical methods in the social sciences, is presented as being intended to achieve the expected results. The validation process of the predictive method will be carried out, taking into account what is proposed by Hernández-Julio et al. [55] to provide methodological rigour to the study.

## 4. Implementation Proposals

The Sequential Predictive Research Design (DISPRE) is a techno-methodological strategy to address complex research problems in the social sciences, which envisages the necessary integration to determine qualitative and quantitative variables’ behaviour. However, it is recognised that data integration can sometimes be problematic due to the object of the research itself, which is ontologically unstable and may not be delimited. Therefore, such messy empirical cuts can be resolved through data di-fractionation using fuzzy logic, allowing for more analytical-predictive density.

This mixed data prediction model is an innovative form of social research that manages to adjust the integration of data through the phases described above, confirming the findings of both types of data, expanding (divergence between data and broadening knowledge) and discordance (inconsistencies between data), revealing multiple phenomena that are entangled in the complex web through the smoothing of discursive triangulation of experts and the application of iterations (Monte Carlo method) that causes one to expand the biases in order to determine and understand them according to their distancing and to consider them as emerging data. It also multiplies bias, increases uncertainty and further entangles the subject and social phenomena. Consequently, through epistemic and methodological pluralism, the different research levels (DISPRE phases) are deepened.

## 5. Future Developments or Limitations

This method is a proposal that has to be validated through application in different fields of social and humanistic sciences. However, predictive analytics shows high accuracy in forecasting eventualities, with fuzzy logic being the most efficient method in data learning. On the other hand, doctoral theses are being advanced with DISPRE for case studies in education sciences, administration and some applications in clinical trials and experimental studies in health sciences.

## Figures and Tables

**Figure 1 entropy-23-00627-f001:**
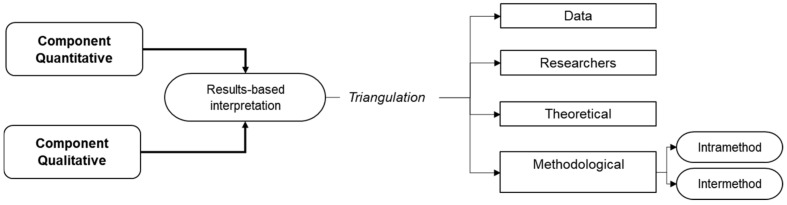
Explanatory sequential research designs, own elaboration (2021) based on Gonzalez-Diaz, et al. [33].

**Figure 2 entropy-23-00627-f002:**
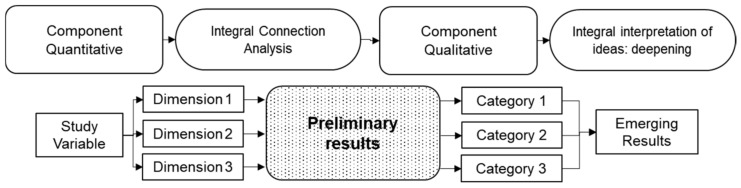
Explanatory sequential research designs, own elaboration (2021) based on Ivankova and Wingo [37].

**Figure 3 entropy-23-00627-f003:**
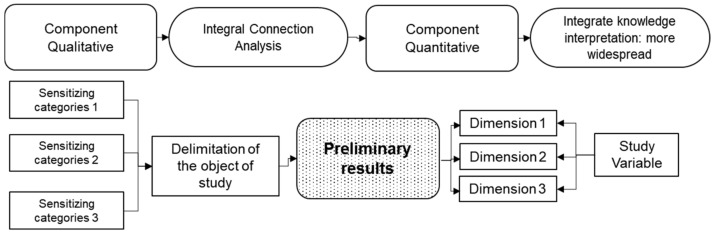
Exploratory sequential research designs, own elaboration (2021) based on Watson et al. [39].

**Figure 4 entropy-23-00627-f004:**
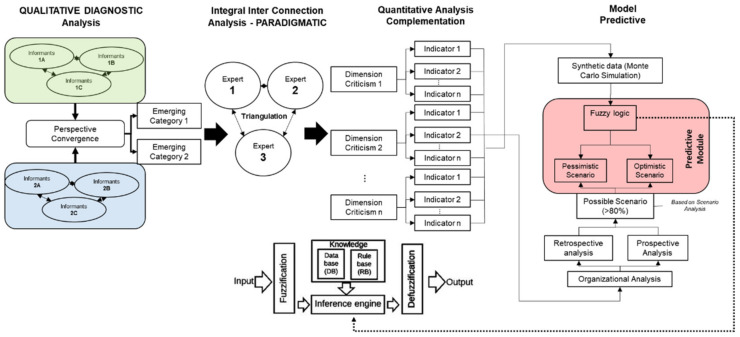
Predictive sequential research design. Own elaboration (2021) based on Cooksey [44,45].

**Figure 5 entropy-23-00627-f005:**
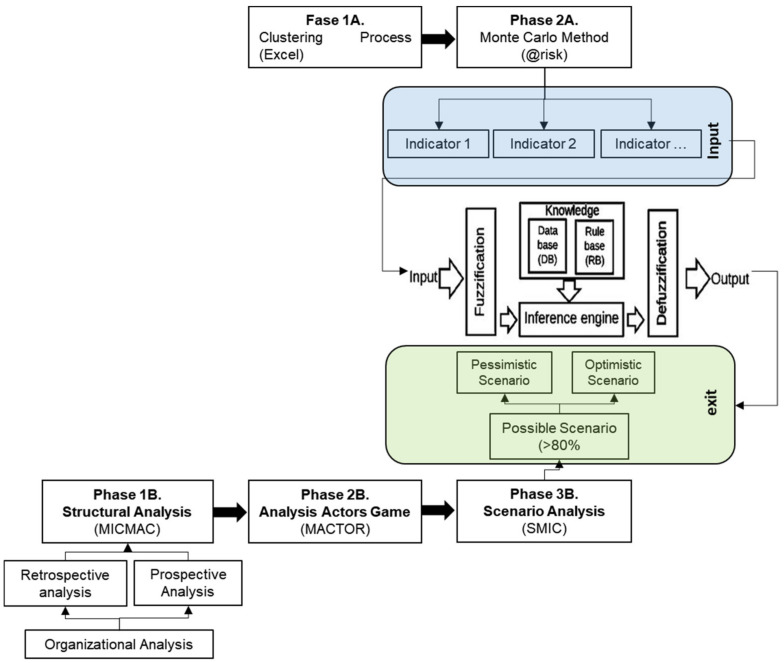
Phases for the configuration of the predictive model, own elaboration (2021).

**Figure 6 entropy-23-00627-f006:**
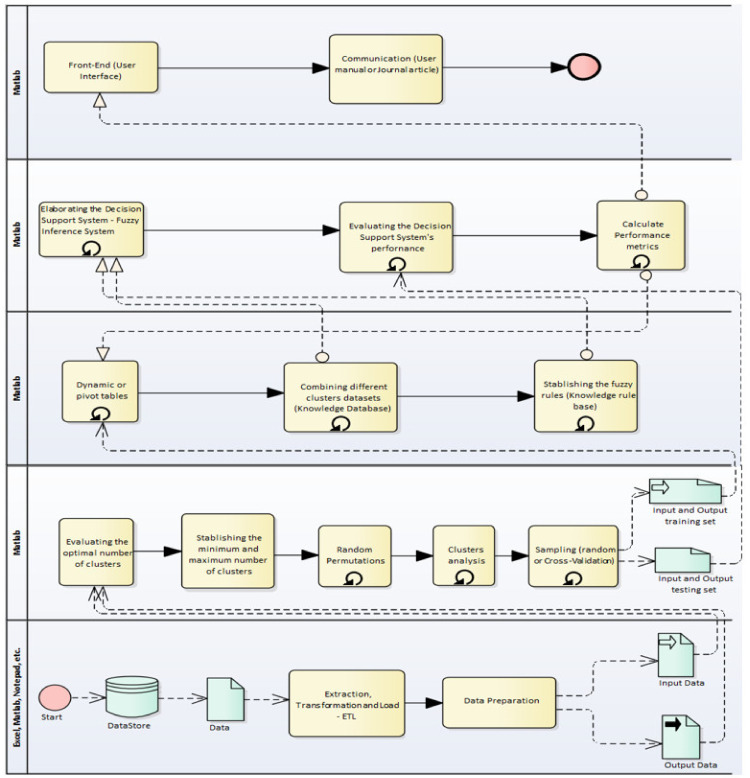
Framework proposed by Hernández-Julio et al. [55].

## Data Availability

Data sharing not applicable.

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
