# Peer review of "Predictive Sequential Research Design to Study Complex Social Phenomena"

_entropy, 2021, doi:10.3390/e23050627_

Round 1

Reviewer 1 Report

In "Predictive Sequential Research Design to study complex social phenomena" authors proposes a predictive sequential research design for social phenomena based on a fuzzy logic as a tool to solve the information biases caused by the investigative diffraction of each methodological approach as a strategy to explain, understand and predict the complexity of the social entity under study.

I have quite enjoyed reading this paper. I find it comprehensive and introducing new esults that might also inspire future research along these lines. For these reasons, I am in favor of revision subject to the following comments.

It would improve the paper if the figure captions would be made more self contained. In addition to what is shown for which parameter values, one could also consider a sentence or two saying what is the main message of each figure.

Also, it would be very useful if the authors would make their source code available as supplementary material. This would promote the usage of the proposed algorithm and allow also others to take advantage of this research, and also to allow them to reproduce the results.

Good recent references for complex social phenomena are Saving human lives: What complexity science and information systems can contribute, J. Stat. Phys. 158, 735-781 (2015) and The social physics collective, Sci. Rep. 9, 16549 (2019). I feel that the subject of complexity in the social sciences should be better captured, and here physics has played a key role. This is quite overlooked, and since Entropy has a strong physics audience, making this link would be important.

If a revision will be granted, I will be happy to review the manuscript again.

Author Response

Distinguished reviewer:
In response to your comments, the respective corrections have been made to the document:

Reviewer's comments 1: The description of some tables has been individualized, since they are correctly explained in each thematic content. Regarding the source code, it should be clarified that it is not unique and should be done in each research depending on the analysis of each researcher. We consider adding the recommended references.

Reviewer 2 Report

Predictive Sequential Research Design to study complex social

phenomena

General 

The authors propose a Predictive Sequential Research Design (DISPRE) for complex social phenomena, which uses fuzzy logic as a tool to solve the information biases caused by the investigative diffraction of each methodological approach as a strategy to capture, explain, understand and predict the intrinsic complexity of the social entity under study.

Main comments

Here are some additional directions and suggestions in which the paper could be improved.

#1 This sentence:

When studying these complex phenomena through a single methodological perspective (quantitative or qualitative), it is usually insufficient, irrational, and illogical. 

is very strong and seems to be disconnected from a broader context. Some elegant and useful descriptions of complex systems do not require ultra-sophisticated perspectives, and still they yield enough insight about a particular field/subject under a particular framework. Consider improving the discussion to support/explain/expand this affirmation, or removing as it may sound offensive or naive.

#2  This sentence in the abstract seems confusing. Consider removing or improving it.

A multi-dimensional reality would be studied in exchange for supposed simple reality, where there are random dynamics almost imperceptible by the systems of current measurements.

#3 Some sentences are very long and confusing to read. For instance, in p. 4 we have:

In this way, it is understood that human beings have in much of their thinking, the use of 153 fuzzy information, which assigns to the elements that make up their ideas a certain degree of belonging, while others are outside the context of their reasoning, i.e. the alternative logic that each subject assumes is different from classical logic, so that the analysis of mixed research methods try to respond in a similar way to the elements of ambiguity and imprecision of information that occur in some social or humanistic research, allowing work with information that is not accurate to be able to perform conventional evaluations.

Another example, pg. 6:

Currently, social sciences are framed in an ontological dimension that assumes a holistic, integral, complex social reality, susceptible of being systematised and valued, with visible, tangible and quantifiable dimensions, as well as non-tangible, non-quantifiable ones epistemologically based on the paradigmatic complementarity Hashimoto and Saavedra [40], which requires being aware of the different purposes of each paradigm, this implies different methodical derivations.

I suggest for the authors to reconsider the writing style for some parts of the paper. Smaller sentences with a closed idea might work well with MDPI Entropy readers.

#4 The paper is well-written, although the authors exaggerate using long sentences, adjectives, and other strange structures (not necessarily wrong). For instance:

Algorithms for analysing multi-criteria decision making and social choice in situations plagued by vagueness (…) 

“Plagued” here is an example of a writing style rather poetical than scientifically precise.

Overall, I think the presentation style is at the edge of this journal. Although the paper exhibit applications in Data Analysis and the work contributes to the field, the writing style looks notably different from what is observed in Entropy. Also, some sentences are tough to read. However, these circumstances do not necessarily disqualify this work for publication. I’d like to politely suggest the authors consider aligning the text style with the journal.

Recommendation

In my opinion, the scientific content of the paper is sound, and the experimental results are described in detail. The treatment of literature seems fair. Furthermore, I believe that enough methodological details are provided for this work to be comprehended and reproduced. Therefore, I recommend that this paper should be accepted for publication after a minor revision if the Editor consider necessary.

Author Response

Distinguished Reviewer:
In response to comments, corrections have been made to the document.

Reviewer 2 observations: Sentence 1 was corrected by placing a less aggressive expression. Similarly, sentence 2 and the very long sentences were separated, summarizing the content of the second observation in more understandable terms.

The term vagueness was changed to accuracy

Hoping to have met your requirements, we are at your disposal for any necessary clarification.

Round 2

Reviewer 1 Report

The authors have revised their manuscript comprehensively and with love to detail. I warmly recommend publication in present form.

Reviewer 2 Report

I think the authors performed relevant improvements in the presentation. I suggest publication of the ms in Entropy in its present form.